# Evolutionary Analysis and Functional Identification of Ancient Brassinosteroid Receptors in *Ceratopteris richardii*

**DOI:** 10.3390/ijms23126795

**Published:** 2022-06-18

**Authors:** Bowen Zheng, Kaixin Xing, Jiaojiao Zhang, Hui Liu, Khawar Ali, Wenjuan Li, Qunwei Bai, Hongyan Ren

**Affiliations:** College of Life Sciences, Shaanxi Normal University, Xi’an 710119, China; zhengbowen365@163.com (B.Z.); xing0818kaixin@163.com (K.X.); zjj22759@163.com (J.Z.); lh13772295316@126.com (H.L.); aali@snnu.edu.cn (K.A.); theandromedagalaxy@163.com (W.L.); baiqw132072@163.com (Q.B.)

**Keywords:** brassinosteroids, BRI1, receptor function, fern, *Ceratopteris richardii*

## Abstract

Phytohormones play an important role in the adaptive evolution of terrestrial plants. Brassinosteroids (BRs) are essential hormones that regulate multiple aspects of plant growth and development in angiosperms, but the presence of BR signaling in non-seed plants such as ferns remains unknown. Here, we found that BR promotes the growth of *Ceratopteris richardii*, while the synthetic inhibitor PCZ inhibits the growth. Using full-length transcriptome sequencing, we identified four BRI1-like receptors. By constructing chimeric receptors, we found that the kinase domains of these four receptors could trigger BR downstream signaling. Further, the extracellular domains of two receptors were functionally interchangeable with that of BRI1. In addition, we identified a co-receptor, CtSERK1, that could phosphorylate with CtBRL2s in vitro. Together, these proved the presence of a receptor complex in *Ceratopteris richardii* that might perceive BR and activate downstream hormone signaling. Our results shed light on the biological and molecular mechanisms of BR signaling in ferns and the role of BR hormone signaling in the adaptive evolution of terrestrial plants.

## 1. Introduction

One hallmark of plant species is the presence of complicated cell communication systems. Unlike animals, plants are sessile and must sense their environment before responding to physiological and morphological changes. These adaptions to the environment primarily rely on a regulatory system of phytohormones [1,2]. From the terrestrial colonization by plants in the Paleozoic to the eventual dominance of angiosperms, the phytohormone regulation systems have continuously evolved, forming various regulation networks [3,4].

Brassinosteroids (BRs) are a class of plant steroid hormones that regulate almost all plant growth and development processes [5,6,7]. In *Arabidopsis thaliana*, mutants that lack BR biosynthesis or their perception are significantly dwarfed and are unable to complete their life cycle. To trigger the BR signal, BRs must first bind to BR receptors. In *Arabidopsis thaliana*, loss-of-function *bri1* mutants are exceptionally dwarfed, similar to the phenotype displayed in the strongest BR biosynthetic mutants, suggesting that BRI1 is the most significant BR receptor. In addition to BRI1, there exist three BRI1-likes (BRLs) [8,9,10]. BRL1 and BRL3 share 80% sequence identity, indicating that BRL1 and BRL3 have recently diverged. Both BRL1 and BRL3 share 43% sequence identity with BRI1, while BRL2 shares 41% sequence identity with BRI1 [11,12]. Therefore, these four BR receptors can be divided into three types: BRI1, BRL1/3 and BRL2 [13]. BRI1 is ubiquitously expressed, while BRL1/3 and BRL2 are restricted to vascular tissues. BRI1 and BRL1/3 can bind to the BRs, whereas BRL2 cannot. Unlike the *bri1* mutant, *brl1/3* and the *brl2* mutants only exhibited visible defects in vascular tissues [8,9,10,11,14].

BRI1-like family receptors (BRLs) belong to the LRR-X (Leucine-Rich Repeats-X) group of receptor-like kinases (RLKs), the largest cell surface receptor family in land plants [15,16]. BRLs consist of an extracellular domain (ECD) rich in leucine repeats (LRRs), a transmembrane domain (TM) and an intracellular kinase domain (KD). There is an intervening sequence called the island domain (ID) between the 5th and 4th LRRs preceding the TM. The hormone-binding site is located in the pocket formed by the ID and the adjacent LRRs [6,8,9,17,18]. BRI1 is negatively regulated by its inhibitor BKI1 (BRI1 kinase inhibitor 1) [19]. When the BR concentration is low, BKI1 binds to BRI1 and inhibits its activity, whereas when the BR concentration is high, BKI1 is phosphorylated and dissociates from BRI1. The co-receptor kinase BAK1 (BRI1-associated kinase 1) is required to activate BRI1, and after detecting BRs, BRI1 and BAK1 form heterodimers and phosphorylate each other sequentially [6,20,21]. The transmission of BR signals from the cell membrane to the transcription factors in the nucleus mainly relies on a series of protein phosphorylation and dephosphorylation reactions. The activated BRI1 consecutively phosphorylates two classes of RLCKs (Receptor-like cytoplasmic kinases), BSK1 (BR-signaling kinases) and CDG1 (Constitutive Differential Growth 1). The BSK and CDG1 then activate the BSU1 (BRI1-Suppressor 1), a PP1-type phosphatase that inactivates the negative regulator BIN2 (Brassinosteroid Insensitive 2), leading to dephosphorylation of the transcription factor BES1/BZR1, thereby inducing the BR response [22,23]. Thus, dephosphorylation of BES1/BZR1 is a marker for the activation of the BR signaling pathway.

Currently, our understanding of BRs biosynthesis and signaling primarily originates from studies in model species, or, to a lesser extent, some angiosperms, which represent a small fraction of extant plant species, many of which have undergone millions of years of evolution [13]. Detailed knowledge of the Arabidopsis BR signaling pathway likely does not represent all other plant lineages. While bioactive BRs and intermediates of biosynthetic pathways are widespread in the plant kingdom [24,25,26,27], and many components of the signaling cascades are conserved throughout the plant kingdom, the critical component of this pathway, the BRI1 receptor, is absent in non-seed plants [13]. Given the presence of physiological responses to BRs in non-seed plants [24,25,26,27], this suggests that BR signaling in non-seed plants differs from that in angiosperms, or that additional BR receptors are involved in their signaling pathways.

Ferns are the second-largest vascular flora, following angiosperms, and can be found from the tropics to the cold temperate biome, and from the lowlands to alpine zones [28,29]. Ferns are the most primitive vascular plants with fossils dating back to the Devonian and have played key roles during plant evolution [30]. Although biological and taxonomical research on ferns is prevalent, relevant genetic research remains in its infancy. Forward genetic studies in ferns are currently difficult, given the lack of genome sequencing and genetic mutants. Therefore, molecular complementation of the respective genetic angiosperm mutants is the optimal alternative for investigating the biological functions of fern genes [31,32]. Our sequence alignments reveal that the BR receptor-like proteins in ferns are more closely related to seed-plant BRL2, which cannot bind BRs. Yet, active BRs and physiological responses to BRs have been detected in ferns [26,27,33], so how BRs function in ferns remains unknown. We found that although the presumed BR receptors in *Ceratopteris richardii* (C-fern) cannot directly complement the dwarf phenotype of Arabidopsis *bri1* mutants, their heterologous expression in Arabidopsis can activate BR responsive genes, indicating that they may have weak BR functions. Current data suggest that different ECDs of RLKs can activate the same ICD, and the same ECD can also activate different ICDs [23,34]. This suggests that we can use chimeric receptors to study the function of various RLKs. Thus, to make the weak BR functions observable, we used molecular engineering techniques to generate different chimeric BR receptors from the presumed BR receptors of C-fern and the BRI1 Arabidopsis receptor by domain swaps [23,35]. Using this approach, we found that the presumed BR receptors in C-fern are functional but less active than BRI1, therefore identifying an ancient BR receptor in C-fern.

## 2. Results

### 2.1. Brassinolide Promotes Growth, Whereas Propiconazole Suppresses Growth in C-Fern

To understand how BRs affect the growth of C-fern, we examined the growth response of C-fern after treatments with epi-brassinolide (epi-BL) and the BR biosynthetic inhibitor, propiconazole (PCZ). We found a positive effect of BRs on the growth and development of C-fern. BRs appeared to highly accelerate the gametophyte development of C-fern. The C-ferns under the 1 µM epi-BL treatment had reached their sporophytic generation concurrent with the gametophytic generation of the control group; however, only a few cordate prothalli were found under the 1 µM PCZ treatment. Seedlings treated with 1 µM epi-BL for 50 days were also larger than the control group, producing longer and wider leaves, while those treated with 1 µM PCZ were the smallest (Figure 1A,B). This suggests that the physiological responses of BRs are conserved in C-fern, leading us to search for homologs of the Arabidopsis BR synthetic gene in the transcriptome sequencing database of C-fern and other available ferns. We found that sequences belonging to the CYP85 clan are present in ferns, with significant similarity to *A. thaliana* CPD, DWF4, and CYP90D1 (Appendix A) [36]. Combined with the study that reported the presence of BRs in pteridophytes, we speculated that there is a BR synthetic pathway in C-fern. We thus detected the expression levels of *CtDWF4* and *CtCPD* in C-fern and found that *CtDWF4* and *CtCPD* were down-regulated while *CtBAS1* was up-regulated in epi-BL-treated plants. In contrast, the opposite was observed in the PCZ treatment (Figure 1C and Appendix A), suggesting that BR signal transduction was activated [5,23,37].

### 2.2. Presence of BRI1-like Receptors in C-Fern

Since BRs act as signaling molecules in ferns, specific receptors are required for their detection and the transition of such signals. While BRI1 and BRL1 can perceive BRs, their orthologs were not found in the transcriptome of ferns (Monilophytes). Instead, we found that all of the presumed BR receptors in ferns were clustered with the BRL2 branch that cannot bind BRs (Figure 1D); as a result, we have named them fBRL2. However, it should be noted that most of the BL interacting residues were conserved between fBRL2 and AtBRL1, especially some of the key residues that bind to BL, such as Phe597, which is identical to AtBRL1, not AtBRL2 (Appendix A). Therefore, fBRL2 may be the bona fide BR receptors.

Four copies of the presumed BR receptors were identified in C-fern, which we named CtBRL2-1, CtBRL2-2, CtBRL2-3 and CtBRL2-4. CtBRL2-1 and CtBRL2-3 share a slightly higher sequence identity, while CtBRL2-2 is closer to CtBRL2-4 (Figure 1A, Appendix A). The expression profile of these receptors showed that the CtBRL2-1 was the highly expressed receptor, especially in sterile fronds, while the expression of CtBRL2-2 was highest in roots (Figure 1E and Appendix A). It is difficult to perform forward genetics studies in C-fern without available mutants, so we chose *Arabidopsis thaliana* mutants for functional complementation experiments [23,38]. We cloned these four presumed BR receptors of C-fern and expressed them under the control of the *AtBRI1* promoter in mutant *bri1-301*. Although the expression of *CtBRL2s* under the *AtBRI1* promoter did not rescue or attenuate the phenotype of *Atbri1* mutants (Figure 2A,E), we detected a weak molecular BR response in *CtBRL2-1* and *CtBRL2-4* transgenic plants (Appendix A), showing that both *CPD* and *DWF4* were slightly down-regulated, while *BAS1* was slightly up-regulated. The dephosphorylation of BES1/BZR1 upon BR treatment is a molecular marker of active BR signaling; thus, we further investigated the accumulation status of phosphorylated vs. dephosphorylated BES1 in transgenic lines after BL treatment. We detected a mild accumulation of dephosphorylated BES1 in *CtBRL2-1* and *CtBRL2-4* transgenic plants (Figure 2F). Together, these results suggest that CtBRL2-1 and CtBRL2-4 cannot effectively perform functions similar to AtBRI1 but can activate molecular markers in BR signals.

### 2.3. The Intracellular Domain of CtBRL2 Could Trigger the BR Signal

Assuming that the inability of CtBRL2 to rescue or attenuate the phenotype of the *Atbri1-301* mutant was due to their weak function, we decided to test whether it can improve their receptor functionality by generating chimeric receptors with AtBRI1 through domain swapping. We fused the BRI1 ECD with the ICD of CtBRL2 to produce an enhanced chimeric receptor, which was introduced into the plants under the control of the BRI1 promoter. We found that all four chimeric receptors of AtBRI1-CtBRL2 could rescue the phenotype of the *Atbri1-301* mutant (Figure 2A), suggesting that the kinase domain of CtBRL2s had similar functionality to AtBRI1, except for the differences in their activity.

To verify whether the AtBRI1-CtBRL2s chimeras participated in the BR signal transduction pathway, we exogenously applied epi-BL and PCZ to these transgenic seedlings. The sensitivity of the chimeric *AtBRI1-CtBRL2s* transgenic plants to BRs was similar to that of WT or transgenics plants expressing BRI1, and PCZ inhibited the hypocotyl elongation of these chimeric transgenic plants in darkness (Appendix A). Additionally, a different level of dephosphorylated BES1 accumulation was detected in *AtBRI1-CtBRL2s* transgenic plants treated with epi-BL (Figure 2E). Further, when compared with WT and *bri1-301* mutants, the expression of other BR-regulated genes was also restored (Appendix A). In conclusion, our results revealed that the chimeric receptor, AtBRI1:CtBRL2s, can functionally replace BR receptors.

We next detected the auto-phosphorylation levels of AtBRI1 and four CtBRL2s using pThr antibody in vitro. The KD domain of each receptor was cloned into pGEX-4T-3 expression vectors with glutathione S-stransferase (GST) tags at the *N* terminal. All purified recombinant vectors exhibited significant autophosphorylation activity under the previously established incubation conditions for AtBRI1 auto-phosphorylation analysis [39,40], but the auto-phosphorylation level of CtBRL2s were all weaker than that of AtBRI1 (Figure 2C). In summary, the ICD of CtBRL2s and AtBRI1 were interchangeable, but the ICD function of CtBRL2s was weaker than that of the BRI1 receptor.

### 2.4. The Extracellular Domains of CtBRL2-1 and CtBRL2-4 Could Replace the Corresponding BRI1 Domains

Given that the ICD of EMS and BRI1 were functionally equivalent, it is reasonable that the ICD of CtBRL2 and AtBRI1 are interchangeable. Further, the ICDs of all BRI1-like receptors in Arabidopsis are interchangeable with the ICD of BRI1 (Figure 3A,B). The difference in receptor function mainly depends on the ECD. In *Arabidopsis thaliana*, the ECD of BRL1 and BRL3 but not AtBRL2 are interchangeable with BRI1 (Appendix A), leading us to explore the ECD function of CtBRL2. We fused the ECD of CtBRL2s and the ICD of AtBRI1, respectively, to construct chimeric receptors of *CtBRL2-1-AtBRI1*, *CtBRL2-2-AtBRI1*, *CtBRL2-3-AtBRI1* and *CtBRL2-4-AtBRI1* and then transferred into the *bri1-301* mutants under the control of the *BRI1* promoter. We found that these four chimeras functioned differently. *CtBRL2-1-AtBRI1* and *CtBRL2-4-AtBRI1* transgenic plants partially rescued the phenotype of *Atbri1-301* mutants with longer petioles and larger rosette diameters. However, the transgenic plants *CtBRL2-2-AtBRI1* and *CtBRL2-3-AtBRI1* did not (Figure 3). Consistently, compared with *Atbri1-301*, the expression levels of *CPD*, *DWF4* and *BAS1* in *CtBRL2-1-AtBRI1* and *CtBRL2-4-AtBRI1* transgenic plants were partially restored, and were also sensitive to exogenous BR treatment (Appendix A). Under BR treatment, dephosphorylated BES1 significantly accumulated in these transgenic plants (Figure 3E), while PCZ inhibited their hypocotyl elongation in the dark (Appendix A).

*bri1-116* is a strong allelic mutant of BRI1 in *Arabidopsis thaliana*, which produces an extremely dwarfed phenotype and cannot complete its life cycle. To further verify the function of the ECD domain of the CtBRL2 receptor, we transferred the chimeras into *bri1-116*. Similar to the previous results, we found that the phenotypes of *CtBRL2-1-AtBRI1* and *CtBRL2-4-AtBRI1* transgenic plants were partially restored (Figure 4A–G). While the plants were still relatively small compared with the WT, the fertility of these plants was significantly restored. We measured the root length of these transgenic plants treated with a high concentration of exogenous eBL. The root lengths of *CtBRL2-1-AtBRI1* and *CtBRL2-4-AtBRI1* transgenic plants were sensitive to eBL (Figure 4H). Consistent with the observed phenotypes, the expression levels of *CPD*, *DWF4* and *BAS1* were also restored (Figure 4J–L). These results indicate that the ECD of CtBRL2-1 and CtBRL2-4 may act as functionally active domains for BR binding but not CtBRL2-2 and CtBRL2-3.

The *cpd* mutant is a strong BR biosynthesis mutant in *Arabidopsis thaliana*, and its phenotype is similar to that of the *bri1* mutant. In order to verify that the phenotypic recovery of *CtBRL2-1-AtBRI1* and *CtBRL2-4-AtBRI1* transgenic plants is dependent on BR signals, we expressed *CtBRL2-1-AtBRI1* and *CtBRL2-4-AtBRI1* under the *35S* promoter in the *cpd* mutant background. We found that these chimeras could not restore the phenotype of the *cpd* mutant (Appendix A), suggesting that the recovery of the *bri1* mutation phenotype by *CtBRL2-1-AtBRI1* and *CtBRL2-4-AtBRI1* were dependent on plant endogenous BR synthesis. In addition, the subcellular localization of CtBRL2s was similar to that of BRI1 and located on the cell membrane (Appendix A). Therefore, we proved that the extracellular domains of CtBRL2-1 and CtBRL2-4 were functionally interchangeable with that of BRI1.

### 2.5. The CDs of CtSERK1 and CtBRL2s Could Transphosphorylate Each Other In Vitro

In addition to BR receptors, activation of the BR signal also depends on its co-receptor BAK1, a small LRR-LRK subfamily with five members named SERK1-SERK5 (Somatic Embryogenesis Receptor Kinase 1 to 5) [41]. The *serk1 serk3 serk4* triple null mutant resembles the phenotype of the loss-of-function *bri1* mutants and is completely insensitive to exogenous BL treatment, indicating that BAK1 is a necessary regulator for the BR signal [42]. Only one SERK family gene was found in the transcriptome of C-fern, which we named CtSERK1. Sequence alignment showed that CtSERK1 contained all of the protein domains similar to AtBAK1. The amino acid sequence of CtSERK1 had 78% identity similarity with that of AtBAK1, and phylogenetic analysis indicates that CtSERK1 is located in the basal branch of the SERK family (Figure 5A and Appendix A). 

To test whether CtBRL2s could interact with CtSERK1, we performed the pull-down assay by obtaining CtSERK1-CD-HIS, CtBRL2-1-CD-GST, CtBRL2-2-CD-GST, CtBRL2-3-CD-GST and CtBRL2-4-CD-GST recombinant proteins. The result showed that the CtSERK1 could interact with all four CtBRL2 (Figure 5B). We next detect whether CtSERK1 and CtBRL2s could transphosphorylate with each other; we performed an in vitro kinase assay. The intracellular kinase domain of CtSERK1 and the four CtBRL2s were cloned into the pGEX-4T-3 expression vectors conferring a GST epitope tag. We also produced a mutated GST-mCtSERK1-CD corresponding to the kinase dead mutant AtBAK1 (K334E). The recombinant fusion protein was purified with glutathione sepharose, and their phosphorylation levels were detected using the pThr antibody under the previously established conditions. The results showed that transphosphorylation between CtSERK1-CD-HIS and CtBRL2s-CD-GST occurs in vitro (Figure 5C,D and Appendix A). The mixture of the same amount of GST-CtSERK1-CD and GST-CtBRL2-CD increased the phosphorylation level CtBRL2s-CD, while the kinase inactivated CtSERK1 (K334E) could not. We inferred that the CtSERK1 in C-fern is a homolog of AtBAK1 in *Arabidopsis thaliana*, can transphosphorylate with CtBRL2, and may mediate BR signaling in C-fern.

## 3. Discussion

The emergence of ferns is an important event in the evolution of terrestrial plants. Ferns evolved vascular tissues, which can support the vertical growth of plants. The prosperity of ferns altered the environment of primitive earth and the evolutionary histories of early animals. Compared with bryophytes, the cellular regulation of ferns is more abundant and complex. For example, although homologs of GID1 and DELLA proteins have emerged in bryophytes, the GA-mediated GID1-DELLA regulatory module has not been developed. In ferns, however, it has formed a GID1-GA-DELLA regulatory module that is very consistent with higher plants [43,44]. The signal of the plant hormone BR studied here is similar, with no homologous BR receptor gene found in the lycopodium of *Selaginella moellendorffii* or the moss *Physcomitrella patens*, which only exists in some lineages such as *Sphagnum mosses*. Thus, BRL genes were presumably present in the common ancestor of mosses and secondarily lost in some lineages [13,45]. The BRL homologous proteins were present in fern *Ceratopteris richardii* as well as all Pteridophyta plants. As a result, we speculate that complete BR signaling is present in ferns. EMS1 and BRI1 signaling share the same downstream signaling pathway, and the main difference lies in their extracellular and ligand-binding domains [23], given that all major transduction components of BR signaling, such as BAK1, BSK1, BSU1, BIN2, are present across land plants. Hence, the most critical molecular event in the origin of BR signaling is the emergence of receptors that can bind to BR.

We conducted BR hormone and synthesis inhibitor experiments in C-fern, revealing that BR promoted the growth of C-fern. In contrast, the synthesis inhibitor inhibited the growth, indicating that C-fern had a physiological response to BR. To identify the presence of BR receptors in C-fern, we performed full-length transcriptome sequencing and identified four *BRL2* homologous genes, which were expressed in different tissues. By using the *BRI1* promoter to drive the *CtBRLs* genes and transferring it into the BRI1 weak mutant *bri1-301*, we found that none of the four genes restored the mutant phenotype. We speculate that this may be due to the weak BR receptor function of CtBRL2, which cannot produce a distinct phenotype. Previous studies have reported that the BR receptor from the gymnosperm *Picea abies* could not restore the Arabidopsis mutant; even the tomato (*Solanum lycopersicum*) SlBRI1 could not restore the *bri1* mutant [32,46]. Considering the long evolutionary distance between Arabidopsis and ferns and the fact that the BR receptor family evolves faster, supporting that the CtBRLs are probably the weak BR receptor, which cannot significantly restore the BR mutants, this supports the hypothesis that the CtBRLs are weak BR receptors, which cannot significantly restore the BR mutants. Furthermore, detection of the accumulation of dephosphorylated BES1 and the expression of endogenous synthesis *CPD*, *DWF4* and degradation genes *BAS1* in transgenic plants supports this hypothesis.

Due to the structural characteristics of the receptor kinase, the intracellular domain and the extracellular domain can be clearly distinguished so that the chimeric receptor can be artificially constructed, and its molecular mechanism can be further studied [35]. To verify whether BRLs can activate the downstream of BR signaling, we utilized the extracellular fusion of *BRI1* and the intracellular fusion of *CtBRLs* to construct a chimeric vector *BRI1-CtBRL*s. This was significantly restored after the transformation into *bri1-301* and promoted the accumulation of dephosphorylated BES1 protein. Thereby revealing that *BRI1-CtBRLs* can activate the BR signal transduction pathway. Conversely, we constructed chimeric receptors *CtBRLs-BRI1* using the extracellular fusion of *CtBRLs* and intracellular fusion of *BRI1* and transferred them to *bri1-301*, finding that two receptors could be significantly recovered by its phenotype. This suggests that CtBRLs may have weak kinase function, and thus the full-length gene cannot be restored, while replacement of the BRI1 kinase domain can significantly restore the *bri1* mutant. In vitro phosphorylation assays revealed that the activity of BRI1 was significantly stronger than that of CtBRLs. Finally, we found that CtSERK1, a homologous gene of the SERKs family, exists in C-fern and can interact with CtBRLs, undergoing mutual phosphorylation. Taken together, we reveal that CtBRLs-CtSERK receptor complexes in C-fern can recognize BR and activate downstream transduction signals (Figure 6).

What role does BR signaling play in the growth and development of C-fern? To answer this question, we needed to knock out BR receptors or key molecular components. Due to the complexity of fern genomes and the immaturity of transformation technology, it remains challenging to create mutants. Recently, it was reported that when the BZR1/BES1 family gene of *Marchantia polymorpha* was knocked out, the growth was significantly reduced, indicating that the BZR1/BES1 component plays an important role in the growth and development of *Marchantia polymorpha* [47]. Based on this, we speculate that BR signaling in ferns plays an important role and that their specific biological functions need to be further explored. In *Arabidopsis thaliana*, BRL2 does not have BR receptor functions [11]. Still, the BRL2 homologous gene revealed in this study exhibits BR receptor functionality, indicating that the BR receptor has been differentiated and sub-functionalized after its emergence in gymnosperms. The BRL1/L3 homologous gene appeared in seed plants, and the powerful BRI1 receptor first appeared in angiosperms recently. The earliest appearance of BRL2, which lost the BR receptor function, evolved to bind other ligands or completely lost the receptor function; this remains unknown. In this study, we revealed the BR receptor complex of C-fern and demonstrated for the first time that BRL2 might act as a functional receptor in non-seed plants. Our study contributes to an in-depth understanding of BR signaling and the evolution of terrestrial plants.

## 4. Materials and Methods

### 4.1. Phylogenetic Analysis

Sequences of BR receptors and BR biosynthases were obtained from phytozome v.13 (https://phytozome-next.jgi.doe.gov/ (accessed on 5 May 2018)), China National GeneBank DataBase (https://db.cngb.org/blast/blast/tblastn/ (accessed on 5 May 2018)) and Ambrorella (http://www.amborella.org (accessed on 5 May 2018)). The sequences obtained were assigned to BRI1 or BRLs according to their similarities to Arabidopsis BR receptors (http://arabidopsis.org/index.jsp (accessed on 5 May 2018)). Multi-sequence alignments were performed with MEGA version 7.0. The identities were calculated by BioEdit v7.1.11. MEGA 7.0 was used to test the optimal model and construct the maximum likelihood tree with 1000 replicates. The tree was visualized using the Figtree v 1.4.4 program.

### 4.2. Plant Materials and Growth Conditions

The *Arabidopsis thaliana* ecotype used in this study was Columbia (Col-0), and the BR mutants *Atbri1-301* and *Atbri1-116* are in Col-0 background. Seeds were placed on half-strength Murashige and Skoog (MS) agar plates after being surface-sterilized [48]. The plates were incubated in the dark at 4 °C for 2 days and were subsequently transferred to a 16 h light/8 h dark cycle at 24 °C for 10 days, then transplanted to soil to obtain mature plants. Four-week-old plants were used for measurements of the width of rosette leaves using the ImageJ software. For measurement of BR-dependent responses, seeds were grown on 1/2 MS medium with or without BL or PCZ in the light or in the dark, and for Western blot analysis, seeds were grown on 1/2 MS medium agar plates under the same conditions for 10 days.

### 4.3. Plant Material and Treatments for Ceratopteris richardii

Spores of *Ceratopteris richardii* were acquired from Paul G. Wolf of the University of Alabama. For measurement of BR-dependent responses, spores were immersed in 2% hypochlorite and 0.1% Tween20 and grown on 1/2 MS medium with or without BL or PCZ to a 16 h light/8 h dark cycle at 24 °C for 30 days and 50 days. Pictures of the plates were taken for measurement of the size using ImageJ software.

### 4.4. Generation of Constructs and Transgenic Plants

We cloned the sequences of four *CtBRL2*, four *AtBRI1-CtBRL2* and four *CtBRL2-AtBRI1* from their respective cDNA by PCR or Overlapping PCR using primers listed in Appendix A and fused them with the *p*CHF3 (kanamycin selection) or *p*CAMBIA1300 (hygromycin B selection) plasmids with GFP flag driven by *BRI1* promoter. All of the above constructs were transferred into the BR mutants *Atbri1-301* or *Atbri1-116* plants with Agrobacterium (GV3101)-mediated transformation by Agrobacterium tumefaciens [49]. The transformants were then screened on half-strength MS with 50 μg/mL kanamycin or 40 μg/mL hygromycin B. At least 10 independent transgenic lines were obtained from each construct.

### 4.5. RT-PCR and qRT-PCR Analysis

Total RNA was extracted from 4-week-old Arabidopsis seedlings or 6-week-old *Ceratopteris richardii* using a Hipure Plant RNA Mini Kit (Magen, Guangzhou, China, R4151-02) according to the manufacturer’s instructions. The cDNA was synthesized from 1 µg of total RNA using a 5× All-In-One RT MasterMix kit (Abm, Zhenjiang, China, Cat# G490). The qRT-PCRs were performed using ChamQTM SYBR qPCR Master Mix (Vazyme, Nanjing, China, Q311-00). The primers used for qRT-PCR and RT-PCR are listed in Appendix A.

### 4.6. In Vitro Kinase Assay

To Measure in vitro kinase activity, the kinase domains of CtBRL2s were cloned into the *p*GEX-4T-3 vector to create GST fusion proteins. The recombinant fusion proteins were expressed in *E. coli* BL21 (DE3) with 0.5 mM isopropyl-β-d-thiogalactoside at 16 °C for 16 h and purified with Glutathione Sepharose 4B (GE Healthcare, Picataway, NJ, USA) according to the manufacturer protocol. Total protein was boiled in 2× SDS-PAGE loading buffer for 5 min and separated on an SDS-PAGE gel. The phosphorylation levels of the target protein were detected by Western blotting with Phospho-Threonine Antibody (1:1000 dilution; CST, Boston, MA, USA, 9381), and the amounts of target protein were quantified by GST antibody (1:5000 dilution, Proteintech, Chicago, IL, USA, #66001-2-Ig).

### 4.7. Pull-Down Assay

The cytoplasmic domains of CtSERK1 were PCR amplified and inserted into the vector *p*ET-28a with a His-tag. The cytoplasmic domains of CtBRL2s were PCR amplified and inserted into the vector *p*GEX-4T-3 with a GST tag. The recombinant fusion proteins were induced with 0.5 mM isopropyl-β-d-thiogalactoside at 16 °C for 16 h. The GST-fused protein and His-fused proteins were purified by Glutathione Sepharose 4B (GE Healthcare, Chicago, IL, USA, 17-0756-01) and Ni-NTA Agarose (QIAGEN, Hilden, Germany, 151032765), respectively. We added CtSERK1-CD-His into GST-CtBRL2s-CD or GST proteins incubated with 20 µL prewashed Glutathione Sepharose 4B beads (GE Healthcare, Piscataway, NJ, USA, 17-0756-01). After 4 h of gentle rocking at 4 °C, we washed the proteins 6–8 times with 50 volumes of His lysis buffer and detected them with His antibody and GST antibody. The reaction of GST with CtSERK1-CD-His was regarded as the negative control.

### 4.8. Protein Extraction and Western Blot Analysis

Seedlings at 16 d old were treated with 100 nM eBL for 90 min and 5 μmol PCZ for 12 h. Total protein was extracted using 2× SDS buffer (100 mM Tris, pH 6.8, 4% [*w*/*v*] SDS, 20% [*v*/*v*] glycerol, 0.2% [*w*/*v*] bromophenol blue, 2% [*v*/*v*] β-Mercaptoethanol) and separated in 10% SDS-PAGE gel at 100 V for 2 h, and then transferred into a nitrocellulose membrane (Millipore, Shanghai, China). The anti-GFP antibodies (1:1000 dilution, TransGen, Beijing, China, HT801) and the BES1 antibodies (1:3000 dilution, kindly provided by J. Li, Lanzhou University, China) were used to detect the GFP fusion protein and the phosphorylation status of BES, respectively. HRP-conjugated secondary antibodies (1:10,000, CWBIO, anti-mouse for GFP, and anti-rabbit for BES1) were used to quantify GFP and BES1 phosphorylation. The signal was visualized using ECL luminous fluid and detected by Tanon Fine Do X6 automatic chemiluminescence image analysis system. Actin was used to determine equal loading.

### 4.9. Statistical Analysis

Statistical analysis was performed using one-way analysis of variance (ANOVA), two-way ANOVA and Turkey’s test, as implemented in GraphPad prism 6.0. Sequence logo was generated using Weblogo online (https://weblogo.berkeley.edu/logo.cgi (accessed on 2 February 2022)).

## Figures and Tables

**Figure 1 ijms-23-06795-f001:**
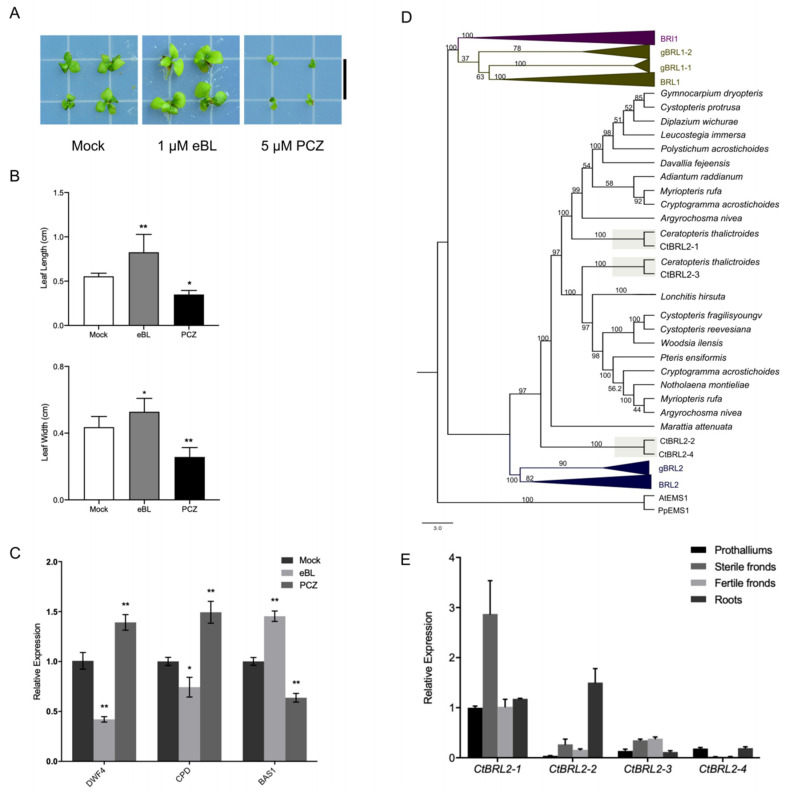
BR could promote the growth of *Ceratopteris richardii*, and four BRL2 homologs were identified in *Ceratopteris richardii*. (**A**) Morphologies of *Ceratopteris richardii* grown on 1/2 MS medium with mock, 1 μM eBL or 5 μM PCZ for 50-day-old. Scale bar = 1.5 cm. (**B**) Leaf length and width of 30-day-old *Ceratopteris richardii* seedlings in 1/2 MS medium treated with mock, 1 μM eBL and 5 μM PCZ. Data are means (± SE), *n* ≥ 10 plants, * *p* < 0.05, ** *p* < 0.01 as one-way ANOVA with a Tukey’s test. (**C**) Quantitative real-time PCR analysis of BR biosynthetic genes *CPD*, *DWF4* and BR catabolic gene *BAS1* of *Ceratopteris richardii* in 6-week-old plants. *n* = 4 biological replicates, * *p* < 0.05, ** *p* < 0.01 as one-way ANOVA with a Tukey’s test. (**D**) Phylogenetic tree of the BRI1 family based on the full-length protein sequences. Sequences were aligned using MUSCLE, and the phylogenetic tree was constructed with MEGA7. The tree was generated using FigTree v.1.4.4 (http://tree.bio.ed.ac.uk/software/figtree/ (accessed on 8 September 2021)). Bootstrap values (in percentages) from 1000 replicates were shown next to the branches. (**E**) Quantitative real-time PCR analysis of four CtBRL2s of *Ceratopteris richardii* in different tissues of 6-week-old plants and prothalliums of 20-day-plant. *n* = 4 biological replicates.

**Figure 2 ijms-23-06795-f002:**
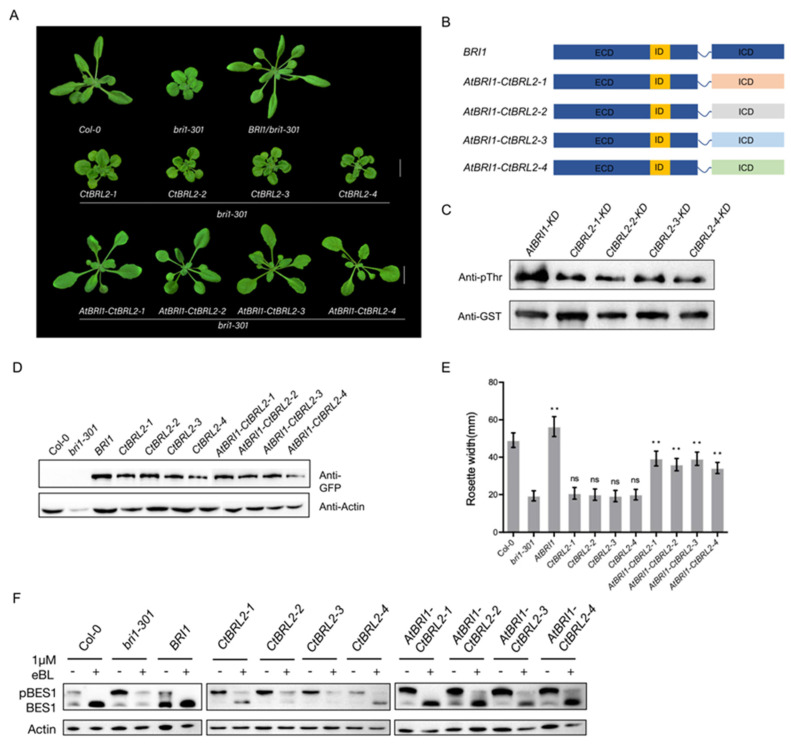
The intracellular kinase domain of four CtBRL2s and BRI1 are interchangeable. (**A**)Phenotypes of 4-week-old transgenic lines expressing *AtBRI1*, *CtBRL2-1*, *CtBRL2-2*, *CtBRL2-3*, *CtBRL2-4*, *AtBRI1-CtBRL2-1*, *AtBRI1-CtBRL2-2*, *AtBRI1-CtBRL2-3* and *AtBRI1-CtBRL2-4* under the *AtBRI1* promoter in *bri1-301*. The scale bar = 1 cm. (**B**) Schematic diagram of the extracellular domain of BRI1 fused with the intracellular kinase domains of four CtBRL2s in *Ceratopteris richardii*. Different receptor fragments are labeled with different colors. (**C**) In vitro kinase activity assay of GST-fused recombinant proteins of the different kinase domains (KDs) labeled as AtBRI1-KD, CtBRL2-1-KD, CtBRL2-2-KD, CtBRL2-3-KD and CtBRL2-4-KD. (**D**) Protein expression levels of the transgenes with GFP tag in the rosette leaves of the corresponding plants shown in (**A**). Actin served as the loading control. (**E**) Quantification of the transgenic lines with the measurement of the rosette leaves diameter of 4-week-old plants, *n* = 15 plants, ** *p* < 0.01, ns, no significance, as one-way ANOVA with a Tukey’s test. (**F**) Phosphorylated BES1 (pBES1) and dephosphorylated BES1 were detected using BES1 antibodies in the extracts of 14-day-old seedlings that were either grown on 1/2 MS medium or treated with 1 μM 24-epibrassinolide (eBL) for 2 h. Actin served as the loading control.

**Figure 3 ijms-23-06795-f003:**
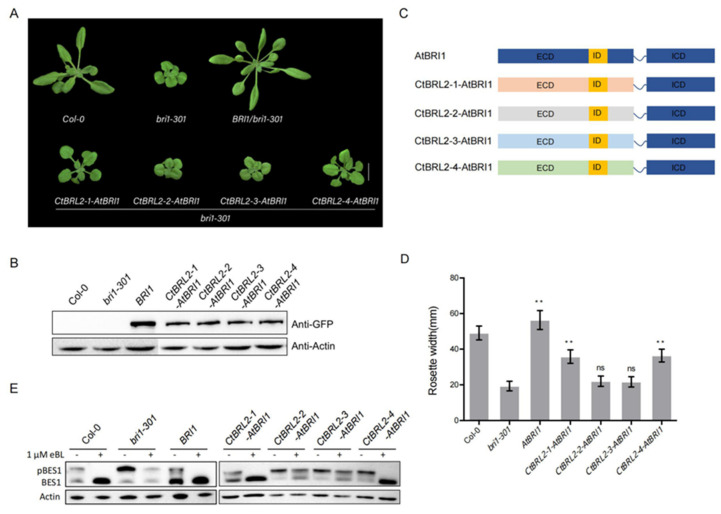
The extracellular domain of CtBRL2-1 and CtBRL2-4 could be interchanged with that of AtBRI1. (**A**) Phenotypes of 4-week-old transgenic lines expressing *AtBRI1*, *CtBRL2-1-AtBRI1*, *CtBRL2-2-AtBRI1*, *CtBRL2-3-AtBRI1* and *CtBRL2-4-AtBRI1* under the *AtBRI1* promoter in *bri1-301*. *CtBRL2-1-AtBRI1* and *CtBRL2-4-AtBRI1* partially rescued the dwarf phenotype of *bri1-301* mutants, while *CtBRL2-2-AtBRI1* and *CtBRL2-3-AtBRI1* could not. The scale bar = 1 cm. (**B**) Protein expression levels of the transgenes with GFP tag in the rosette leaves of the corresponding plants shown in (**A**). Actin served as the loading control. (**C**) Schematic diagram of the chimeric receptors of CtBRL2-1-AtBRI1, CtBRL2-2-AtBRI1, CtBRL2-3-AtBRI1 and CtBRL2-4-AtBRI1 labeled in different colors, respectively. (**D**) Quantification of the transgenic lines with the measurement of the rosette leaves diameter of 4-week-old plants, *n* = 15, ** *p* < 0.01, ns, no significance, as one-way ANOVA with a Tukey’s test. (**E**) Phosphorylated BES1 (pBES1) and dephosphorylated BES1 were detected using BES1 antibodies in the extracts of 14-day-old seedlings that were either grown on 1/2 MS medium or treated with 1 μM 24-epibrassinolide (eBL) for 2 h before the preparation of the extracts. Actin served as the loading control.

**Figure 4 ijms-23-06795-f004:**
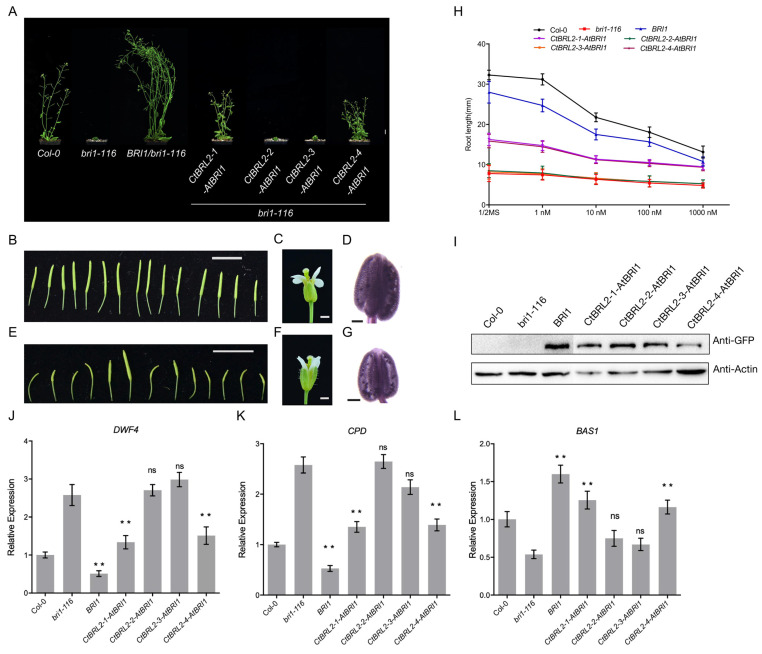
CtBRL2-1-AtBRI1 and CtBRL2-4-AtBRI1 can function as BR receptors to restore the phenotype of *bri1-116* mutant. (**A**) Phenotypes of 7-week-old transgenic lines expressing *BRI1*, *CtBRL2-1-AtBRI1*, *CtBRL2-2-AtBRI1*, *CtBRL2-3-AtBRI1*, *CtBRL2-4-AtBRI1* under the *BRI1* promoter in *bir1-116*. Scale bar = 1 cm. (**B**,**E**) All siliques in a branch of 7-week-old plants of Col-0, *CtBRL2-1-AtBRI1* in *bri1-116*, respectively. Scale bars = 1 cm. (**C**,**F**) Mature flower of Col-0, *CtBRL2-1-AtBRI1* in *bri1-116*. Scale bars = 1 mm. (**D**,**G**) Alexander staining of pollen grains in mature anthers of Col-0, *CtBRL2-1-AtBRI1* in *bri1-116* showing the fertility phenotypes of *CtBRL2-1-AtBRI1* in *bri1-116*. (**H**) Root lengths of 5-day-old seedlings grown in 1/2 MS medium supplemented with different concentrations of eBL. Numbers indicate means (± SE), *n* > 10. (**I**) Protein expression levels of the transgenes with GFP tag in the rosette leaves of the corresponding plants as shown in (**A**). Actin served as the loading control. (**J**–**L**) A similar response was found in BR-regulated genes in the transgenic lines of *CtBRL2-1-AtBRI1*, *CtBRL2-4-AtBRI1* and *BRI1*. Quantitative real-time PCR analysis of BR biosynthetic genes *DWF4* and *CPD* or BR catabolic gene *BAS1* in 4-week-old plants. *n* = 5, ** *p* < 0.01, ns, no significance (one-way ANOVA with Tukey’s test).

**Figure 5 ijms-23-06795-f005:**
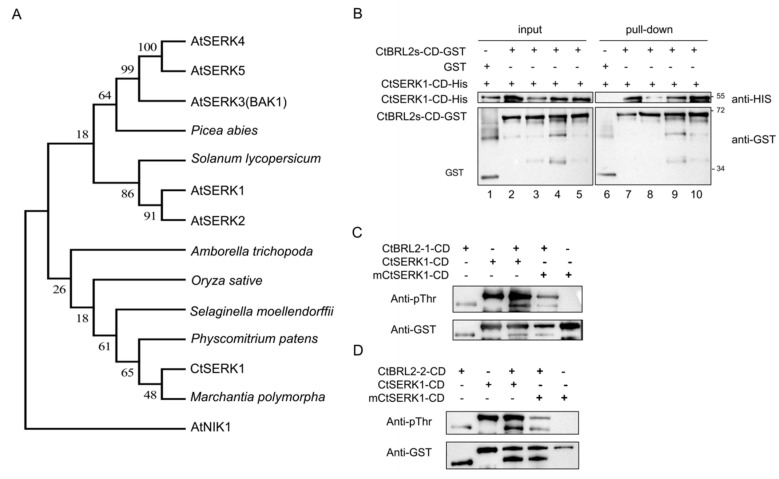
The CtSERK1 works as the co-receptor of CtBRL2s. (**A**) Phylogenetic tree of the SERK family based on the full-length protein sequences. The tree was generated with MEGA7 and visualized by the FigTree v.1.4.4 (http://tree.bio.ed.ac.uk/software/figtree/ (accessed on 12 December 2021)). Bootstrap values (in percentages) from 1000 replicates were shown next to the branches. Accessions numbers for the sequences used for alignments are as follows: AtSERK1 (AT1G71830), AtSERK2 (AT1G34210), AtSERK3 (AT4G33430), AtSERK4 (AT2G13790), AtSERK5 (AT2G13800), AtNIK1 (AT5G16000), Picea abies (MA_10428962g0010), Oryza sative (LOC_Os08g07760), Physcomitrium patens (Pp3c8_3330V3.1), Marchantia polymorpha (Mapoly0068s0069), Solanum lycopersicum (Solyc04g072570.2.1), Amborella trichopoda (AmTr_v1.0_scaffold00010.295), Selaginella moellendorffii (v1.0|268032). (**B**) Pull-down assays showed four CtBRL2s could interact with CtSERK1. GST and CtBRL2s-CD-GST were immobilized on GST beads, incubated with CtSERK1-CD-His protein, and immunoblot analysis was detected with His antibody and GST antibody. Each lane represents an independent reaction: lane 1, CtSERK1-CD-HIS + GST; lane 2, CtSERK1-CD-HIS + GST-CtBRL2-1-CD; lane 3, CtSERK1-CD-HIS + GST-CtBRL2-2-CD; lane 4, CtSERK1-CD-HIS + GST-CtBRL2-3-CD; lane 5, CtSERK1-CD-HIS + GST-CtBRL2-4-CD; lane 6, CtSERK1-CD-HIS + GST; lane 7, CtSERK1-CD-HIS + GST-CtBRL2-1-CD; lane 8, CtSERK1-CD-HIS + GST-CtBRL2-2-CD; lane 9, CtSERK1-CD-HIS + GST-CtBRL2-3-CD; lane 10, CtSERK1-CD-HIS + GST-CtBRL2-4-CD. (**C**,**D**) CtSERK1-CD and CtBRL2-CD can both autophosphorylate each other. The kinase assays were performed using CtSERK1-CD-GST and CtBRL2-CD-GST. Top panel, phosphorylation analyzed by pThr antibody. Bottom panel, the GST served as the loading control.

**Figure 6 ijms-23-06795-f006:**
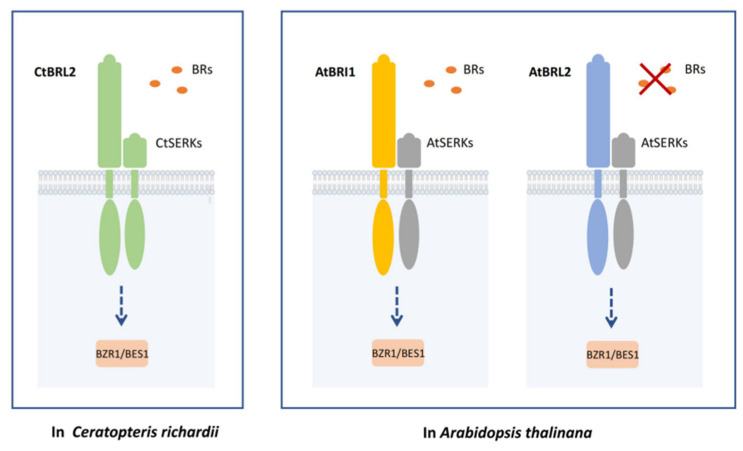
A proposed model for the BR signaling in *Ceratopteris richardii* and *Arabidopsis thaliana*.

## Data Availability

The data or material of this study are available from the corresponding author, H.Y.R., upon reasonable request.

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
