# Peer review of "Evolutionary Analysis and Functional Identification of Ancient Brassinosteroid Receptors in Ceratopteris richardii"

_ijms, 2022, doi:10.3390/ijms23126795_

Round 1

Reviewer 1 Report

In present study “Evolutionary analysis and functional identification of ancient brassinosteroids receptors in Ceratopteris richardii” Zheng and colleagues explore possible molecular mechanisms of brassinosteroid (BR) signaling in fern Ceratopteris and existence of BR signaling module in this model organism. Although I find their experimental approach well-structured and logical and presented data solid, I believe that authors should tone down their interpretation of results and claim that BR signaling module exists in Ceratopteris. Nevertheless, this manuscript contains valuable information for BR research field and should be published after some restructuring. Although this study, in my opinion, does not decisively answer the question whether BR signaling module exists in Ceratopteris, it still provides new insights into this important evolutionary question.

Comments:

1. Identification of CtDWF4 and CtCPD is an important finding. Nevertheless, authors did not try to directly detect presence of BRs in Ceratopteris. I understand that this is challenging, but without this experiment, claim for BR signaling complex in ferns is only indirectly supported. This should be emphasized in the text.

2. Molecular docking combined with Alphafold protein structure prediction can be performed on CtBRL2s to test if brassinolide or similar molecules can bind to island domain of the receptor.

3. Line 158. Changes in BES1 phosphorylation are minor and not quantified. I am not convinced that this receptor can trigger BR signaling in Arabidopsis. In addition, plant phenotype of bri1-301 is not rescued.

4. -The extracellular domains of CtBRL2-1 and CtBRL2-4 could replace the corresponding BRI1 domains-

This section of paper is the strongest part of the study. Especially introduction of chimeric receptor in null bri1-116 mutant. I find it very probable that CtBRL2 ECD can bind brassinolide or some related molecule. I suggest that authors test fern responses to some other bioactive brassinosteroids (e.g. castasterone or typhasteol) since it is not known if Ceratopteris can synthetize brassinolide. As an ultimate proof of BL binding to CtBRL2 authors should consider performing grating-coupled interferometry to determine dynamics of brassinosteroid(s) binding to CtBRL2, similar to Hohmann et al., 2018.

Author Response

Referee: 1

Comments to the Author
In present study “Evolutionary analysis and functional identification of ancient brassinosteroids receptors in Ceratopteris richardii” Zheng and colleagues explore possible molecular mechanisms of brassinosteroid (BR) signaling in fern Ceratopteris and existence of BR signaling module in this model organism. Although I find their experimental approach well-structured and logical and presented data solid, I believe that authors should tone down their interpretation of results and claim that BR signaling module exists in Ceratopteris. Nevertheless, this manuscript contains valuable information for BR research field and should be published after some restructuring. Although this study, in my opinion, does not decisively answer the question whether BR signaling module exists in Ceratopteris, it still provides new insights into this important evolutionary question.

Comments:                                                                                                                          

  1. Identification of CtDWF4 and CtCPD is an important finding. Nevertheless, authors did not try to directly detect presence of BRs in Ceratopteris. I understand that this is challenging, but without this experiment, claim for BR signaling complex in ferns is only indirectly supported. This should be emphasized in the text.

Response: We truly appreciate the nice comments and are particularly thankful for the constructive suggestions that make this manuscript much better.

We strongly agree with the reviewer. The concentration of BR in plants is extremely low, and direct detection is very difficult. Due to the limited resources, we demonstrated the BR signaling module in ferns solely through indirect genetic approaches. To support our claim, we have also cited a study that reported the BR detection in ferns and made appropriate modifications to the conclusion section.

  1. Molecular docking combined with Alphafold protein structure prediction can be performed on CtBRL2s to test if brassinolide or similar molecules can bind to island domain of the receptor.

Response: This indeed is a good way to test the interaction of island domain with BRs. The problem we encountered is that the structure prediction of the island domain is based on BRI1 and BRL1/BRL3. Apparently, the BR receptor is closer to BRL2, and the crystal structure data of BRL2 is currently not available. Predicting the structure of the BR receptor in Ceratopteris is difficult and the results would not be very reliable. In the future, we plan to study and detect the direct interaction between BR and BRL2 from Ceratopteris. But of course, this would also be a very challenging experiment.

  1. Line 158. Changes in BES1 phosphorylation are minor and not quantified. I am not convinced that this receptor can trigger BR signaling in Arabidopsis. In addition, plant phenotype of bri1-301 is not rescued.

Response: We agree with the reviewer that the accumulation of dephosphorylation BES1 is mild and not quantified and relying solely on BES1 phosphorylation experiments does not provide sufficient evidence to claim that the full-length genes for these two receptors can activate BR signaling, especially the fact that bri1-301 mutant phenotypes cannot be restored. Therefore, we refrained and modified the statement.

  1. The extracellular domains of CtBRL2-1 and CtBRL2-4 could replace the corresponding BRI1 domains.

This section of paper is the strongest part of the study. Especially introduction of chimeric receptor in null bri1-116 mutant. I find it very probable that CtBRL2 ECD can bind brassinolide or some related molecule. I suggest that authors test fern responses to some other bioactive brassinosteroids (e.g. castasterone or typhasteol) since it is not known if Ceratopteris can synthetize brassinolide. As an ultimate proof of BL binding to CtBRL2 authors should consider performing grating-coupled interferometry to determine dynamics of brassinosteroid(s) binding to CtBRL2, similar to Hohmann et al., 2018.

Response: We appreciate the reviewer’s very good question and suggestion. At this point, we tried to demonstrate that CtBRL2 is able to directly bind BR and further conduct experiments on the kinetics of CtBRL2 binding to BR. Given the time constraint provided and limited resources, we would save this study for the future reports. Instead, we reconstruct our paper to solely focus on BR-related responses in fern.

The BR synthesis of Ceratopteris shows an interesting fact, such as CYP85A1 and CPD are the key enzymes of BR synthesis, Arabidopsis CPD cannot restore CYP85A1, but we recently found that CtCPD of Ceratopteris can not only restore the Arabidopsis cpd mutant, but also the cyp85a1 mutant, suggesting that the synthetic genes of Ceratopteris may be multifunctional. Of course, more evidence is still needed.

Thanks again for your careful review!

Reviewer 2 Report

I would suggest to commit the next considerations:

1. Given that the lack of data of the endogenous content of BRs in ferns, I would recommend to include the reference Fernández et al. 2021. Front. Plant Sci., 12 November 2021 | https://doi.org/10.3389/fpls.2021.718932

2. Please, revise the meaning of the sentences in lines 346 and 356.

3. Please, add the reference of Murashige and Skoog 1962.

4. In the list of references, please remove Caño-Delgado as it is repeated.

Author Response

Referee: 2

Comments and Suggestions for Authors

I would suggest to commit the next considerations:

  1. Given that the lack of data of the endogenous content of BRs in ferns, I would recommend to include the reference Fernández et al. 2021. Front. Plant Sci., 12 November 2021 | https://doi.org/10.3389/fpls.2021.718932

Response: Thanks for your careful review!

We agree with the reviewer. The concentration of BR in plants is extremely low, and direct detection is very difficult. This study lacks the detection information of the endogenous BR content of Ceratopteris, and we have cited the study as the reviewer asked, which is very suitable to support our conclusion.

  1. Please, revise the meaning of the sentences in lines 346 and 356.

Response: We have revised the whole paragraph to ensure ease of understanding and smoothness.

  1. Please, add the reference of Murashige and Skoog 1962.

Response: We added that reference as the reviewer requested.

  1. In the list of references, please remove Caño-Delgado as it is repeated.

Response: We apologise for the careless mistake and have made changes.  Thank you!
